# Mesopelagic Species and Their Potential Contribution to Food and Feed Security—A Case Study from Norway

**DOI:** 10.3390/foods9030344

**Published:** 2020-03-16

**Authors:** Anita R. Alvheim, Marian Kjellevold, Espen Strand, Monica Sanden, Martin Wiech

**Affiliations:** Institute of Marine Research, P.O. Box 1870, Nordnes, NO-5817 Bergen, Norway; anitaroyneberg@hotmail.com (A.R.A.); marian.kjellevold@hi.no (M.K.); Espen.Strand@hi.no (E.S.); Monica.Sanden@hi.no (M.S.)

**Keywords:** mesopelagic, nutrients, *Benthosema glaciale*, *Maurolicus muelleri*, trace elements, minerals, fatty acids, vitamin A, vitamin D

## Abstract

The projected increase in global population will demand a major increase in global food production. There is a need for more biomass from the ocean as future food and feed, preferentially from lower trophic levels. In this study, we estimated the mesopelagic biomass in three Norwegian fjords. We analyzed the nutrient composition in six of the most abundant mesopelagic species and evaluated their potential contribution to food and feed security. The six species make up a large part of the mesopelagic biomass in deep Norwegian fjords. Several of the analyzed mesopelagic species, especially the fish species *Benthosema glaciale* and *Maurolicus muelleri*, were nutrient dense, containing a high level of vitamin A1, calcium, selenium, iodine, eicopentaenoic acid (EPA), docosahexaenoic acid (DHA) and cetoleic acid. We were able to show that mesopelagic species, whose genus or family are found to be widespread and numerous around the globe, are nutrient dense sources of micronutrients and marine-based ingredients and may contribute significantly to global food and feed security.

## 1. Introduction

One of the greatest societal challenges of the twenty-first century is to secure sufficient and nutritious food for all in a sustainable manner [1]. Currently, two billion people suffer from vitamin and mineral deficiencies, especially in vitamin A, iron and zinc [2]. Micronutrient deficiencies, known as hidden hunger, can severely affect health and development and in some cases lead to irreversible effects [3]. As many as 842 million people suffer from chronic hunger, meaning an insufficient amount of food for an active life [2] and two billion people consume excess calories [4]. By 2050, the global population is projected to rise to 9.6 billion, demanding a 60% increase in global food production [5]. Food from the ocean has a large potential to meet this need, and contribute to food security due to its highly nutritious nature [6], containing essential vitamins, minerals, long-chain omega-3 fatty acids, and other nutrients not found in plant-based or terrestrial animal sources [7,8,9]. Fish also enhance the bioavailability of minerals like iron and zinc [10,11,12,13] especially from cereal- and legume-based meals.

A recent report concluded that better ocean and fisheries management globally could increase catches by 20% compared with current levels [6]. Whether the future increased demand for marine food production from fisheries and aquaculture can be met will largely depend on the effects of climate change mitigation, the global implementation of ecosystem-based fisheries management [14], and the aquaculture’s capacity to expand in a sustainable way [15]. A shift in diets from terrestrial, animal-based protein towards ocean-based options may reduce the triple burden of malnutrition, and contribute significantly to climate change mitigation [16].

Fish account for about 20% of the global intake of animal protein and for almost 7% of all protein consumed by humans [17,18]. The existing supply of marine raw materials cannot meet the nutritional demand for human consumption nor feed production needed for the aquaculture industry to grow [17]. Thus, there is a need to use more biomass from the ocean as future food and feed, preferentially from lower trophic levels, such as organisms from the mesopelagic zone [19]. The mesopelagic zone, stretching from 200 to 1000 meters depth, comprises about 60% of the planet`s surface and 20% of the ocean volume, constituting a large part of the total biosphere. The total amount of mesopelagic fish biomass is suggested to be in the order of 10,000 million tons globally—equivalent to 100 times the annual catch of traditional fisheries [17,20,21,22]. Mesopelagic organisms have the potential to become a major contributor to global nutrition and can play an important part in national and global bioeconomy if exploited in a sustainable manner. However, mesopelagic species remain one of the least investigated biomasses in terms of distribution, abundance, fishing methods and product development. There is little information on the nutrient content or nutritional value of mesopelagic species, which is needed to evaluate its potential as a novel food or feed resource.

This paper contributes with novel data on the nutritional content of six of the most abundant mesopelagic species in fjords of western Norway, with the genera or families to which they belong being found to be widespread and numerous in mesopelagic ecosystems all around the globe [20,23,24]. We also evaluate their potential contribution to food and feed security.

## 2. Materials and Methods

### 2.1. Biological Material

Two species of mesopelagic fish, the glacier lanternfish (*Benthosema glaciale*) and the silvery lightfish (*Maurolicus muelleri*), the decapod *Eusergestes arcticus*, the decapod genus *Pasiphaea* (comprising *P. multidentata*, *P. sivado* and *P. tarda*), the euphausiid Northern krill (*Meganyctiphanes norvegica*) and the scyphozoan helmet jellyfish (*Periphylla periphylla*) were harvested in three different fjords of the Norwegian west coast; Osterfjorden, Bjørnafjorden and Boknafjorden (Figure 1). Specimens were caught in mesopelagic trawls between December 5th and 9th, 2018, onboard the research vessel “Johan Hjort”. The macroplankton trawls used are pelagic otter trawls with equal sized mesh throughout the length of the trawl with a mount opening of either ~35 m^2^ [25,26] or a larger version with ~350 m^2^ opening. Both trawls were equipped with sensors for the continuous in situ measurement of the trawl’s opening width, height and flow speed. The trawl sensor data were later used for the calculation of total water filtered and in combination with trawl catches, the mean density of species/group in the water column were calculated. All trawl hauls used for later calculation of biomass density were oblique hauls from the surface down to around 350 m, thus filtering equal amounts of water as a function of depth. All trawl catches were sorted and identified to the highest possible taxonomic level before being weighted separately (Appendix A). For large catches, only a subsample of the remaining mixed catch was sorted after large and uncommon specimens had been removed. After sorting and weighting, all common species of fish and crustaceans were measured for length.

At least one pooled sample was prepared for each species/genus from each location for later chemical analysis. For *B. glaciale*, *M. muelleri* and *M. norvegica* from Osterfjorden, samples were divided into different size classes, and, for *B. glaciale,* also according to sex. *P. periphylla* was only sampled from catches from Osterfjorden (*n* = 12) and Bjørnafjorden (*n* = 10) and the total wet weight was used as size measurement. A quarter of each *P. periphylla* individual was used in the composite sample.

All pooled samples were homogenized directly after catch and distributed into different tubes for separate analysis and stored frozen at −20 °C until December 17th, 2018. Thereafter, all samples were stored at −80 °C until analysis.

### 2.2. Analytical Methods

Analyses of nutrients of the composite sample were performed at the Institute of Marine Research (IMR) in Bergen, Norway. All analyses, except for iron and fatty acids, were performed using accredited methods according to ISO 17025:2005. The laboratory participates in national and international proficiency tests to secure trueness and establish measurements uncertainty of the methods. Certified reference materials (CRM) were analyzed to test for accuracy and all values presented were within the accepted range of the analyses. For all methods, a sample blank and a quality control sample (QC) with a known composition and concentration of target analyte were included in each series. The methods were regularly verified by participation in inter-laboratory proficiency tests, or by analyzing certified reference material (CRM), where such exist.

The limit of detection (LOD) is the lowest level at which the method is able to detect the substance, while the limit of quantification (LOQ) is the lowest level for a reliable quantitative measurement. The LOQ for the analytical methods used for the nutrients presented in this paper can be found in Reksten et al. (submitted to Journal of Food Composition and Analysis).

Protein (crude protein) was determined by burning the sample material in pure oxygen gas in a combustion tube at 830 °C. Nitrogen (N) was detected with a thermal conductivity detector according to the accredited method AOAC Official Methods of Analysis [27]. Nitrogen content was calculated from an estimated average of 16% N per 100 gram of protein using the formula; N g/100g − 6.25 = protein g/100g.

Fat (crude fat) was extracted with ethyl acetate and filtered before the solvent evaporated and fat residue weighted. The method is standardized as a Norwegian Standard, NS 9402.

Fatty acids were analyzed on a HP-7890A gas chromatograph (Agilent, Santa Clara, CA, USA) with a flame ionization detector (GC-FIS) as described in [28] with the nonadecanoic acid (19:0) as internal standard. For this, 2.5 M dry HCl in methanol was used as a methylation agent. The fatty acids methyl esters (FAME) were extracted using 2 × 2 mL hexane. Several of the samples contained wax esters and the hexane extracts were added nonadecanol (19:0 alk) as internal standard and fatty alcohols were separated using solid phase column (500 mg aminopropyl-SPE, Supelco). The FAME fraction was eluted with 3 mL hexane + 2 mL hexane:ethyl acetate 9:1 v/v) and the fatty alcohols were eluted with 4 mL chloroform. The extracted hexane was diluted or concentrated to obtain a suitable chromatographic response. One microliter was injected splitless with an injection temperature of 280 °C. A 25 m × 0.25 mm fused silica capillary, coated with polyethylene-glycol of 0.25 μm film thickness, CP-Wax 52 CB (Varian-Chrompack, Middelburg, The Netherlands) was used. Helium was used as mobile phase at 1 mL/min for 45 min and then increased to 3 mL/min for 30 min. The temperature of the flame ionization detector was set at 300 °C. The oven temperature was programmed to hold at 90 °C for 2 min, then from 90 to 165 °C at 30 °C/min and then to 240 °C at 2.5 °C/min and held there for 35 min. Fifty-seven FAME peaks and fifteen fatty alcohols peaks were selected in the chromatograms, and identified by comparing retention times with a FAME standard (GLC-463 from Nu-Chek Prep. Elysian, MN, USA) and fatty alcohol standard (GLC-33-36A from Nu-Chek Prep. Elysian, MN, USA), and retention index maps and mass spectral libraries (GC-MS) (http://www.chrombox.org/home/) performed under the same chromatographic conditions as the GC-FID [29]. Chromatographic peak areas were corrected by empirical response factors calculated from the areas of the GLC-463 mixture. The chromatograms were integrated using the EZChrom Elite software (Agilent Technologies).

Vitamin A_1_ (sum all-trans retinol and 13-, 11-, 9 cis retinol) was determined by an analytical high-performance liquid chromatography (HPLC) (normal phase) using a PDA detector (Photo Diode Array) (1260 Infinity, Agilent). The sample was saponified and the unsaponified material was extracted. Retinol content was calculated by external calibration (standard curve) [30].

The sample for determination of vitamin D_3_ was saponified and the unsaponifiable material was extracted and purified on a preparative HPLC column. The fraction containing D_2_ (ergocalciferol) and D_3_ (cholecalciferol) was pooled (normal phase). This fraction was injected into a HPLC column (reverse phase). Vitamin D_2_/D_3_ was determined by an UV detector. The content of vitamin D_3_ was calculated using an internal standard (vitamin D_2_) [31].

Selenium (Se), zinc (Zn), iron (Fe), calcium (Ca), potassium (K), magnesium (Mg), phosphorus (P) and sodium (Na) were determined by Inductively Coupled Plasma-Mass Spectrometry (iCapQ ICPMS, Thermofisher Scientific, Waltham, MA, USA) equipped with an autosampler (FAST SC-4Q DX, Elemental Scientific, Omaha, NE, USA) after wet digestion in a microwave oven (UltraWave of UltraClave, Milestone, Sorisole, Italy) as previously described [32] with some modifications. The elements were quantified using an external standard curve in addition to an internal standard [33]. Three slightly different methods were applied: (1) for Ca, Na, K, Mg, and P using scandium (Sc) as the internal standard, (2) for Zn and Se, rhodium (Rh) was used as the internal standard, and 3) tellurium (Te) was used as the internal standard for iodine (I). For the determination of iodine, the sample preparation was a basic extraction with tetramethylammonium hydroxide (TMAH) before ICP-MS analysis.

### 2.3. Data Management and Presentation of Data

All analytical data were exported from Laboratory Information Management Systems (LIMS) to Microsoft Excel Office 365 ProPlus for calculation of means and standard deviation (SD). Data are presented as means ± SD per 100g wet weight of several composite samples of each species and reported to units of expressions and rounding procedures as advised in the FAO guidelines “Food composite data” [34]. For values <LOQ, for further calculations, the respective LOQ was divided by 2, as suggested by Helsel [35]. Vitamin A components are presented as µg/100 g of the vitamin A isomers retinol (sum of 13-, 11-, and 9-cis and all-trans retinol (A_1_)) and 3.4 didehydro-all-trans retinol (A_2_). Values for vitamin A_1_ were included in the calculations, while values for vitamin A_2_ were excluded due to the small amount present and the reduced biological activity of dehydroretinaol isomers [36]. For vitamin A_1_, 1 µg = 1 RE (retinol equivalent). Vitamin D is presented as the amount of vitamin D_3_ present in sample, as the amount of vitamin D_2_ is considered negligible in fish [37]. Nutrients are presented by species and the mean values from different fjords are merged.

### 2.4. Biomass Density

In order to estimate the biomass density of the species in the fjord, the total catch of each species was divided by the total amount of water that had passed through the trawl:$${\mathrm{Biomass}\;\mathrm{density}}_{\mathrm{species}}=\frac{{\mathrm{Biomass}\;\mathrm{in}\;\mathrm{Trawl}}_{\mathrm{species}}}{{\mathrm{Trawl}}_{\mathrm{area}}\times {\;\mathrm{Trawl}}_{\mathrm{speed}}\times {\;\mathrm{Trawl}}_{\mathrm{time}}}$$

Biomass densities calculated from oblique trawl hauls only report the average biomass in the fjord down to the deepest point of the trawl’s depth profile, and hence do not take into account the fact that mesopelagic organisms tend to aggregate in diel vertical migrating layers [38]. Consequently, these layers will likely have a much higher than average density and a future fishery targeting the layers would catch more biomass per volume trawled than reported here. The estimate is based on 12 trawls in Bjørnafjorden and 5 trawls in Osterfjorden. The catches from Boknafjorden cannot be considered quantitative, thus no species composition was attempted for this location.

#### Nutritional Potential

The catch composition (Figure 2) shows the average value of species-specific biomass densities from each trawl haul in the two fjords, as well as the mean of the two fjords for the investigated species. The latter values were used to calculate the amount of protein, fat and selected micronutrients (g) present per km^3^ ($A_{nutrient}^{})$ fjord as:$$A_{nutrient}^{}=\overset{n}{\underset{species=1}{\sum }}\left(Conc_{species}^{nutrient}\times Biomass\;density_{species}^{}\right)$$
where $Conc_{species}^{nutrient}$ is the concentration of a specific nutrient in a specific species (g/kg) and $Biomass\;density_{species}^{}$ represents the average trawl catch of each species in the two fjords (kg/km^3^).

To calculate the potential concentrations of the different nutrients after processing into an oil and protein fraction the following assumptions were made; (1) the processing would result in a protein and oil fraction similar to what we gained from our chemical analysis for total fat and protein. (2) 100 % of all elements were following the protein fraction. (3) 100 % of itamin A was following the oil fraction.

Under these assumptions, the concentration of nutrients in the protein or oil fraction $Conc_{oil/protein}^{nutrient}$ was calculates as
Concoilproteinnutrient=ConcspeciesnutrientConcspeciesoilprotein
where $Conc_{species}^{oil/protein\;}$ is the concentration of total fat and protein in a specific specie.

The nutritional potential (potential daily doses of recommended intake (RI)) of nutrients per km^3^ fjord from the mesopelagic species was based on the Nordic Nutrition Recommendations [39] for women for the selected nutrients; iodine, calcium, iron, zinc, selenium and vitamin A_1_, and the calculated amount of the nutrients in each species (Appendix A):Potential dosesdaily=ConcspeciesnutrientRI

## 3. Results

Novel data on the nutrient composition in six mesopelagic species from Norwegian fjords and their potential for global food and feed security are presented here. We also compare our findings with the nutrient content of other protein sources, and Blue whiting (*Micromesistius poutassou*), one of the main commercial industry fishes used to produce fishmeal and fish oil in Norway.

### 3.1. Sample Characteristics

This study included six mesopelagic species; two fish species, *B. glaciale* and *M. muelleri*, three shellfish/crustacea *M. norvegica, Pasiphaea spp*. and *E. arcticus* and one jellyfish *Periphylla perihylla*, sampled in three fjords in western Norway; Osterfjorden, Bjørnafjorden and Boknafjorden, December 2018 (Figure 1).

The six species presented here make up a large and continuous part of the mesopelagic biomass density in deep Norwegian fjords (Figure 2).

An overview of the species sampled is given in Table 1.

### 3.2. Nutrient Dense Mesopelagic Species

The protein content was similar for all species except the jellyfish *P. periphylla*, and comparable to Blue whiting, whereas fat content varied greatly (Table 2).

The fish species *B. glaciale* and *M. muelleri* contained high levels of vitamin A_1_ (retinol) (Table 3). Vitamin A_2_ was only detected in *B. glaciale* and *M. muelleri* at 26.0 ± 7.5 µg/100g and 27.8 ± 7.2 µg/100g, respectively (mean ± SD). In the other species, vitamin A_2_ was under the limit of quantification (LOQ) (<0.5 µg) and vitamin D was under LOQ for all species (data not shown). All species, except the jellyfish *P. periphylla*, contained high levels of calcium and selenium, whereas iodine content varied considerably between the species (Table 3).

High amounts of monounsaturated fatty acids were found in all species, mainly 18:1n-9, 20:1 n-9, 20:1n-11 (gadoleic acid) and 22:1n-1 (cetoleic acid), and eicopentaenoic acid (EPA) and docosahexaenoic acid (DHA) (Table 4). The content of DHA was higher than EPA in all mesopelagic species, in contrast to commercial fish oil (Table 4).

*B. glaciale* and *E. arcticus* contained considerable amounts of wax esters (long-chain fatty acid alcohols esterified to long-chain fatty acids), of which 16:0, 20:1 and 22:1 constituted the major fatty acids, respectively (Appendix A).

Most of the nutrients analyzed are similar to, or higher than, that of commonly consumed fish fillets such as farmed Atlantic salmon (*Salmo salar*), Atlantic cod (*Gadus morhua*) and meat (Table 5), and Blue whiting commonly used to produce fishmeal and fish oil for the aquaculture industry.

### 3.3. Potential Contribution to Combat Micronutrient Deficiency

The mesopelagic species investigated here are nutrient dense for several important micronutrients relevant for global food and feed security. According to recommended intake (RI) for adult women, consuming 50 gram of the shrimp *E. arcticus* and krill *M. norvegica* (raw) will provide >30% of RI for iodine, calcium and selenium, whereas 50 gram of the shrimp *Pasiphaea* and the two fish species *B. glaciale* and *M. muelleri* will provide >30% of RI for calcium and selenium (Table 5). The estimates of biomass density in Oster- and Bjørnafjorden of the six species presented here give an average of 312 tonnes WW/km^3^ of jellies, and 26 tonnes WW/km^3^ of the other five species (Figure 2). Based on these estimates, 1 km^3^ of fjord contain huge potential amounts of protein, fat and several micronutrients (Appendix A). One cubic kilometer has the potential to provide about 169,000 daily “doses” of the recommended intake of iodine, 591,000 doses RI of selenium, 31,800 doses RI of iron and 87,300 doses RI of zinc (Table 5).

The mesopelagic fish species presented here are comparable to sprat, a small dietary fish, both in nutritional content (Table 5) and appearance (Appendix A).

### 3.4. Potential Contribution to Aquaculture Compared to Commercially Available Marine Feed Ingredients

Fish meal is the main source of macro- and microminerals of all feed ingredients used in commercial aquaculture diets. The mineral content in the mesopelagic species presented here (Table 3) are comparable to industry fishes such as Blue whiting and capelin (*Mallotus villosus*, https://sjomatdata.hi.no/#search/). Our calculations of minerals in the protein fraction (Appendix A) show that the mesopelagic species without jellies are as mineral dense as common commercially available fish meal [41]. Today, iron, zinc and selenium are generally added as premixes to commercial Norwegian salmonid diets due to the high content of terrestrial feed ingredients. However, for novel marine feed ingredients and their potential contribution to aquaculture, in addition to their nutrient content, nutrient bioavailability should always be evaluated.

## 4. Discussion

Here, the nutrient composition of six mesopelagic species from Norwegian fjords is discussed and their contribution to global food and feed security evaluated. The worldwide biomass of mesopelagic fish based on trawl catches was estimated at about 1000 million tons [20,42]. However, acoustic surveys indicate this to be an underestimate of at least one order of magnitude [21], possibly due to trawl avoidance [22]. Mesopelagic communities are observed acoustically worldwide as deep-scattering layers, performing varying degrees of diel vertical migrations [38]. The fish species *B. glaciale* and *M. muelleri* belong to the families Myctophidae and Sternoptychidae, respectively, which both have a worldwide distribution comprising more than 300 different species combined [42]. In addition to mesopelagic fishes, the mesopelagic community remains poorly understood [43,44], but comprise significant quantities of pelagic shrimps, euphausiids, squids and jellies [45,46,47]. A future fishery targeting the mesopelagic layers might be a mixed fishery where the catch will comprise of different species making up the mesopelagic community, and knowledge about the community composition and the nutritional contents will be needed.

Alleviating different forms of hunger effectively requires political commitment and strategies that go beyond conventional health and nutrition systems, and this has been on the agenda since early 1990s. Micronutrient malnutrition affects health, but also impacts socioeconomic development, learning abilities and productivity [48]. Food-based strategies including diet diversity (promoting foods that are naturally rich in micronutrients) is one of the most sustainable solutions [49]. To reduce the prevalence of hidden hunger and the triple burden of malnutrition, multiple sectors, such as agriculture, health, nutrition and the environment, should be involved, aiming to improve people’s diets in a sustainable manner [1]. Sustainability studies regarding seafood often lack consideration of either nutritional or health aspects of the products in question [50]. Most seafoods are preferable from a climate perspective compared to pork and especially beef [50]. Characterizing the nutritional content of the mesopelagic community will add valuable data to better understand the relative nutritional benefits of foods from the ocean. Such information can enable the transition towards more healthy and sustainable diets and ensure food and feed security, and work towards achieving several of the UN Sustainable Developmental Goals to end hunger and secure safe, nutritious and sufficient food (SDG 2), ensure good health (SDG 3) and conserve and sustainably use the oceans, seas and marine resources for sustainable development (SDG 14).

The high content of several important minerals in the mesopelagic species (Table 3) may be naturally attributed to various parts of the specimen present. Both fish species were analyzed with head and viscera, which may explain the high levels of vitamin A_1_ (Table 3). In *Amblypharyngodon mola*, a small indigenous fish commonly consumed whole in Bangladesh, 90% of vitamin A was found in the eyes and viscera [51,52]. Vitamin A deficiency is a leading cause of preventable blindness in children and is a public health problem in many African and South-East Asian countries [53]. A serving of 50 gram of the two lantern fish species *B. glaciale* and *M. muelleri* (raw) provide 117% and 73% of recommended intake of vitamin A in women, respectively (Table 5). The amount of these fish species in the two investigated Norwegian fjords has the potential to provide approximately 348,000 daily doses of vitamin A for women per km^3^. Lanternfish are found in various locations around the world [54,55]. If other mesopelagic fish species contain similar levels of vitamin A as *B. glaciale* and *M. muelleri*, these species may be an effective way to supplement the diet with vitamin A in low-income countries. All data presented here are from whole, raw specimens. Processing might affect the amount of micronutrients [56,57]. Thus, further studies are needed to investigate the impact of processing on micronutrient content in the mesopelagic species presented here.

Micronutrient deficiency in vulnerable stages of life can affect both physical and mental health. Iodine deficiency is one the main causes of impaired cognitive development in children [58]. In Norway [59,60] and other European countries [61], pregnant women have suboptimal iodine status, which may affect infant development [62,63,64]. Inadequate iodine intake was observed in various subgroups such as women at childbearing age, the elderly and vegans [65] and pregnant women [60,65]. Also, the risk of suboptimal usual iodine intakes among children and adolescents varies with age, sex, maternal educational level and area of residence [66]. Consuming 50 g of several of the mesopelagic species (raw) may contribute to 15%–40% of the recommended intake of iodine (Table 5).

All the mesopelagic species, except the jellyfish *Periphylla*, contained high amounts of calcium. Calcium from small, soft-boned species commonly eaten whole is as efficiently absorbed as from milk, making them an important source of calcium, especially in developing countries where milk intake is low and small fish are part of the everyday diet [67,68].

The level of iron in the mesopelagic species presented here would potentially contribute to 1%–7% of RI in adult women (Table 5). Dietary iron from locally available small marine fish contributes to food-based strategies to reduce the risk of iron deficiency in rural Cambodians [57]. Small nutrient-dense fish are important for food and nutrition security and could contribute to fighting iron deficiency, especially in vulnerable groups [67,68]. Fish also enhance the bioavailability of iron and zinc, especially from cereal- and legume-based meals [10,11,12,13]. Iron deficiency is the most common nutritional disorder worldwide [69], and anemia due to iron deficiency is associated with significantly lower scores in cognitive and educational achievement tests in school-aged children, and lower work productivity in adults [69]. Based on biomass estimations, there is a large potential for iron and zinc (31,800 and 87,300 daily doses of RI per km^3^, respectively, Table 5) from mesopelagic species in the investigated fjords.

In contrast to commercially available fish oils, the mesopelagic species contained higher amounts of DHA compared to EPA (Table 4), also found in the lantern fish *B. pterotum* from the Gulf of Oman [55] and krill (*Euphausia superba*) [56]. The amount of monounsaturated fatty acids was high in the lantern fish species (Table 5), of which cetoleic acid (22:1n-11) was especially high. This fatty acid is also found in high amounts in herring [70,71], capelin and sand eels [72]. A herring diet and herring oil counteracted the negative metabolic effects in rats induced by a high-fat, high-sugar diet, probably due to the lipid composition being rich in EPA, DHA, cetoleic acid and gadoleic acid (20:1n-11) [71]. A recent study found that supplementation with saury oil, a fish oil high in gadoleic acid and cetoleic acid, improved plasma lipids in healthy subjects, indicating that there are other nutritional components besides EPA and DHA in fish which are important for cardiovascular health [73].

The high content of wax esters in *B. glaciale* and *E. arcticus* may affect the lipid profile of these species (Table 4). Compared to triacylglycerol, wax esters are considered to be less bioaccessible due to poorer digestibility in both mammals [74] and Atlantic salmon [75]. However, fatty alcohols may be oxidized [76] and hydrolyzed [77] in the digestive tract of rodents and can therefore not completely be ignored as a nutrient. It is not known if it is the digestion, absorption, elongation or oxidation of wax esters that may regulate their nutritional value [77].

In addition to a potential as a nutrient-dense food source (Appendix A), the large biomass of mesopelagic species may also be used as feed ingredients in aquaculture. Already, studies have shown that cetoleic acid, which is especially high in lantern fish species, stimulates the capacity of human and salmon cells to produce EPA and DHA, and enhance the retention of EPA and DHA in Atlantic salmon [72]. One key challenge for sustainable aquaculture development is sufficient feed ingredients that can produce healthy and robust fish [17,78,79]. Currently, Norwegian farmed salmon are fed more than 70% plant feed ingredients on average [80], with similar levels in Chile, Canada and Australia, while Scotland salmon diets contain slightly higher levels of marine feed ingredients [81]. This blue green shift has changed the dietary supply and availability of marine lipid soluble nutrients [82] and micro-minerals [83,84,85]. The potential of using oil or meal from the mesopelagic biomass could be one of the solutions to secure sufficient and nutritious feed for the aquaculture industry. The use of mesopelagic biomass as a feed ingredient in aquaculture would depend on several factors, such as the level of legacy undesirables [86], the nutrient bioavailability and the nutrient composition in processed oil and meal products. The mesopelagic species presented here, and mesopelagic fish especially, are comparable, and even more nutrient dense for some nutrients such as iodine and calcium (Table 3), fat content (Table 2), and several fatty acids such as gadoleic acid, cetoleic acid, EPA and DHA (Table 4), compared with the Blue whiting commonly used for the production of fishmeal and fish oil. Yearly catches of Blue whiting are approximately 400,000 tons in Norway (https://www.fiskeridir.no/Yrkesfiske/Tall-og-analyse/Fangst-og-kvoter/Fangst/Fangst-fordelt-paa-art).

Small fish consumed whole including the head and viscera are already part of multiple food cultures [52,87,88], which probably also will apply to most mesopelagic species being of a small size. Food-based approaches to tackle micronutrient deficiencies improve the overall quality of a diet compared to micronutrient supplementation, which is unlikely to ensure a sustainable improvement of diets worldwide [1]. The data presented here are important in order to make fisheries policy more nutrition sensitive [89], and we need reliable and high-quality representative data regarding nutrient content of food from the ocean. Although the Nordic countries benefit from a safe and sufficient food supply [90], unhealthy diets are a leading risk factor for poor health. Nordic food systems have not been able to reduce the negative development in non-communicable nutrition-related diseases and put pressure on the environment domestically and abroad [89]. Norway and many other countries have food-based dietary guidelines adapted to their nutrition status, food availability, culinary culture and eating habits [91], but we need to make the guidelines more nutrition-sensitive. Mesopelagic species may contribute to achieve nutrition-sensitive food system as nutrient-dense food sources either directly as food or indirectly as feed ingredients. As a next step, it has to be evaluated how this new resource can be exploited in a sustainable manner.

All data presented in this paper were analyzed at a national reference laboratory using accredited methods, except for fatty acids analysis. The analytical data reported here are an important contribution to the insight into nutrient content of mesopelagic species that may be an important food and feed source in the future. Since the here applied crude protein method is somewhat uncertain due to the assumption that all measured nitrogen comes from protein and a standard amino acid composition, in future studies, the amino acid profile should be considered for calculating the true protein content [92,93]. The species presented here represent the majority of the mesopelagic biomass in three fjords in western Norway. However, many species, such as lanternfish and krill, are also found worldwide, making the data relevant in a global perspective.

In the present work, the nutrient composition is presented at the species level. Catches of mesopelagic species can vary tremendously in terms of species composition. Our data show a large variation in nutrient composition of the species. Accordingly, the nutrient profiles of the catches will also vary. The commercial mesopelagic fishery is still in an early stage and, presently, it cannot be foreseen what the main use of the resource will be. There might be a targeted fishery for some more valuable species, or it might be catching a bulk biomass for processing. Either way, species-specific data is particularly valuable to predict the nutrient profile of a catch, given some information on the species composition of the catch.

We are aware that mesopelagic fish can contain anti-nutrients that may lead to unfavorable and potentially adverse effects for farmed fish and humans [86,94], and levels of contaminants in the species presented here ([86,94,95], Wiech et al. in prep) that may limit the use for human consumption and as feed ingredients, but that is outside the scope of this paper. The data presented here are based on samples from a single research cruise and the nutrient content of the species may vary according to factors including season, inter-annual fluctuations, fish size and fishing equipment. Olsen et al. [95], however, found no pronounced effect of season on different nutrient levels, while size affected the fat content.

## 5. Conclusions

In this study, six mesopelagic species from three fjords in Western Norway were investigated for nutritional composition and evaluated in terms of food and feed security. Small fish eaten whole have potential as a nutrient-dense animal source, both contributing with micronutrients and enhancing bioavailability from vegetable sources. Several of the mesopelagic species were nutrient-dense, containing high levels of vitamin A_1_, calcium, selenium, iodine, EPA and DHA. Due to a large biomass, mesopelagic species, as marine-based proteins and a source of micronutrients, may contribute significantly to global food and feed security, if harvested and managed sustainably.

## Figures and Tables

**Figure 1 foods-09-00344-f001:**
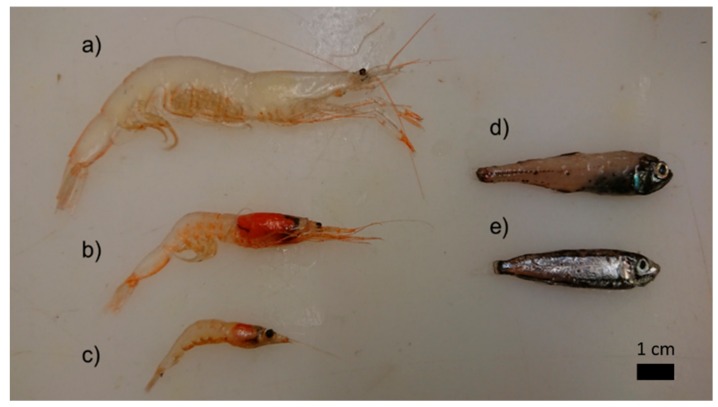
Mesopelagic species in Norwegian fjords. The shrimps (**a**) *Pasiphaea sp.* and (**b**) *Eusergestes arcticus*, the krill (**c**) *Meganyctyphanes norvegica*, and the fish species (**d**) *Benthosema glaciale* and (**e**) *Maurolicus muelleri* caught at a cruise in Osterfjorden, Bjørnafjorden and Boknafjorden on the Norwegian west coast in December 2018.

**Figure 2 foods-09-00344-f002:**
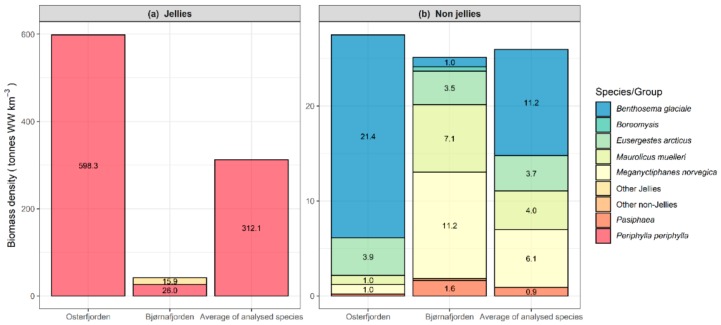
Biomass density of mesopelagic species in Norwegian fjords. Biomass density of mesopelagic species/groups in Osterfjorden and Bjørnafjorden in December 2018 from oblique trawls with macroplankton trawls with either 35 or 350 m^2^ opening area. The fjords contained most of the same species; however, their contribution to the total ecosystem varied greatly. (**a**) The jellyfish *Periphylla periphylla* in Osterfjorden and Bjørnafjorden and an average of the 2 fjords. (**b**) Mesopelagic species, without jellies, in Osterfjorden and Bjørnafjorden, and an average of the 5 other species mentioned in this paper.

**Table 1 foods-09-00344-t001:** Overview of analyzed samples. Number of composite samples including number of specimens in each sample and the average length or weight of the specimens (mean ± SD) from Osterfjord, Bjørnafjord and Boknafjord (December 2018) is given.

Location	Species	Classification	Composite Samples (n)	Specimens per Composite Sample	Average Length (mm)
Osterfjorden	*Benthosema glaciale*	Pisces	4	27	62.6 ± 2.3
				135	50 ± 2.8
				135	48.8 ± 2.9
				>50	23.9 ± 5.6
	*Maurolicus muelleri*	Pisces	2	>50	23.6 ± 2.6
				>50	44.5 ± 3.9
	*Meganyctiphanes norvegica*	Crustacea	2	>50	17.2 ± 2.1
				>50	33.2 ± 2.3
	*Pasiphaea sp.*	Crustacea	1	>50	70.1 ± 10.6
	*Eusergestes arcticus*	Crustacea	2	>50	32.2 ± 5.2
				>50	26.4 ± 4.8
Boknafjorden	*Benthosema glaciale*	Pisces	1	>50	52.7 ± 6
	*Maurolicus muelleri*	Pisces	1	>50	48.5 ± 5.9
	*M. norvegica* (Northern krill)	Crustacea	1	>50	33.4 ± 2.4
	*Pasiphaea sp.*	Crustacea	1	>50	82 ± 8
	*Eusergestes arcticus*	Crustacea	1	>50	50.3 ± 10.3
Bjørnafjorden	*Benthosema glaciale*	Pisces	1	83	41.4 ± 10.5
	*Maurolicus muelleri*	Pisces	1	>50	36.6 ± 9.4
	*M. norvegica* (Northern krill)	Crustacea	1	>50	30.3 ± 5.1
	*Pasiphaea sp.*	Crustacea	1	>50	49.3 ± 19.6
	*Eusergestes arcticus*	Crustacea	1	>50	43.6 ± 7.2
					**Weight (g)**
Osterfjorden	*Periphylla periphylla* (Helmet jellyfish)	Cnidaria	1	12	575 ± 446
Bjørnafjorden	*Periphylla periphylla* (Helmet jellyfish)	Cnidaria	1	10	952 ± 293

**Table 2 foods-09-00344-t002:** Analytical wet weight-based values for protein, total fat and dry matter in six mesopelagic species caught in three fjords in western Norway, and Blue whiting for comparison.

Species	*n*	Protein g/100g (min–max)	Total Fat g/100g (min–max)	Dry Matter % (min–max)
*Benthosema glaciale* (Glacier lantern fish)	7	14.0 ± 0.5 (13.5–14.6)	13.7 ± 3.7 (6.1–16.0)	30.8 ± 3.9 (22.0–33.7)
*Maurolicus muelleri* (Silvery lightfish)	4	12.3 ± 0.4 (11.9–12.7)	17.8 ± 8.1 (7.1–24.7)	33.3 ± 8.1 (23.0–41.2)
*Meganyctyphanes norvegica* (Northern krill)	4	15.5 ± 0.9 (14.8–16.8)	5.5 ± 0.6 (4.9–5.9)	24.0 ± 1.9 (21.3–25.3)
*Pasiphaea sp.*	3	14.1 ± 4.6 (42–50)	5.4 ± 2.7 (3.3–8.4)	21.7 ± 5.1 (15.9–24.1)
*Eusergestes arcticus*	4	15.5 ± 0.5 (14.9–15.9)	9.4 ± 3.1 (4.9–12.1)	27.5 ± 3.6 (22.3–30.7)
*Periphylla periphylla* (Helmet jellyfish)	2	0.95 (0.90–1.00)	0.45 (0.34–0.56)	4.82 (4.76–4.87)
*Micromesistius poutassou ** (Blue whiting)	10	16.1 (15.5–17.1)	3.9 (2.9–5.8)	20.8 (18.4–22.9)

Data are expressed as mean ± standard deviation, and minimum and maximum values. *n* = number of composite samples. * Measurements on individual samples, data from (https://sjomatdata.hi.no/#search/)

**Table 3 foods-09-00344-t003:** Analytical wet weight-based values of vitamin A1, iodine and selected minerals in six mesopelagic species caught in three fjords in western Norway, and Blue whiting (*M. poutassou*) for comparison.

Species	*n*	Vitamin A1 µg/100g	Iodine µg/100g (min–max)	Calcium mg/100g (min–max)	Potassium mg/100g (min–max)	Magnesium mg/100g (min–max)	Phosphorus mg/100g (min–max)	Sodium mg/100g (min–max)	Selenium µg/100g (min–max)	Zinc mg/100g (min–max)	Iron mg/100g (min–max)
*B. glaciale*	7	1633 ± 356 (1300–2300)	43 ± 6 (30–49)	500 ± 47 (420–550)	258 ± 51 (160–300)	67 ± 12 (52–89)	383 ± 60 (280–440)	385 ± 108 (300–600)	61 ± 9 (47–72)	0.8 ± 0.1 (0.7–1.0)	1.08 ± 0.44 (0.61–1.83)
*Maurolicus muelleri*	4	1020 ± 395 (480–1400)	27 ± 14 (16–47)	543 ± 60 (480–600)	227 ± 6 (220–230)	61 ± 8 (54–70)	400 ± 10 (390–410)	380 ± 69 (340–460)	44 ± 8 (34–52)	1.1 ± 0.1 (1.1–1.2)	1.56 ± 0.05 (1.50–1.60)
*M. norvegica*	4	63.3 ± 15.3 (50.0–80.0)	119 ± 42 (85–180)	658 ± 57 (590–730)	358 ± 33 (320–390)	163 ± 13 (150–180)	368 ± 22 (340–390)	495 ± 124 (360–660)	101 ± 41 (71–160)	1.0 ± 0.1 (0.9–1.1)	2.15–1.39 (0.98–4.00)
*Pasiphaea sp.*	3	11.0 ± 1.0 (10.0–12.0)	46 ± 4 (42–50)	633 ± 211 (410–830)	283 ± 110 (160–370)	83 ± 29 (53–110)	333 ± 119 (200–430)	337 ± 107 (220–430)	43 ± 21 (23–65)	0.9 ± 0.3 (0.6–1.1)	0.39 ± 0.30 (0.19–0.74)
*Eusergestes arcticus*	4	34.5 ± 29.6 (6.0–60.0)	117 ± 6 (110–120)	532 ± 88 (460 -660)	358 ± 22 (300–420)	378 ± 22 (350–400)	377 ± 22 (350–400)	363 ± 51 (300–420)	52 ± 17 (38–76)	1.8 ± 0.9 (1.0–3.1)	0.32 ± 0.11 (0.23–0.45)
*Periphylla periphylla*	2	0.3 (0.15–0.45)	2.3 (2–2.5)	43 (42–44)	83 (80–86)	105 (110–110)	12.3 (9.6–15.0)	1000 (1000–1000)	3.9 (3.4–4.4)	0.1 (0.1–0.1)	0.04 (0.04–0.05)
*M. * poutassou*	10	2370 (1000–4500)	23 (19–34)	429 (198–785)	264 (233–282)	64 (55–72)	309 (222–517)	425 (373–466)	62 (60–64)	1.1 (1.0–1.2)	1.75 (1.50–2.00)

Data are expressed as mean ± standard deviation, and minimum and maximum values in brackets. *n* = number of composite samples. * Measurements on individual samples, data from https://sjomatdata.hi.no/#search/.

**Table 4 foods-09-00344-t004:** Absolute and relative values of selected fatty acids in 6 mesopelagic species caught in three fjords in western Norway, and for comparison Blue whiting (*Micromesistius poutassou)* and fish oil intended for aquaculture feed production (mean ± SD).

	*B. glaciale* g/100 g ww min–max % (*n* = 8)	*Maurolicus muelleri* g/100 g ww min–max % (*n* = 4)	*M. norvegica* g/100 g ww min–max % (*n* = 4)	*Pasiphaea sp.* g/100 g ww min–max % (*n* = 3)	*E. arcticus* g/100 g ww min–max % (*n* = 4)	*P. periphylla* g/100 g ww min–max % (*n* = 2)	*M. poutassou ** g/100g ww min–max (*n* = 10)	Fish oil ** % (n=10)
Amount FA (g/100 g sample weight)	6.8 ± 1.7	14.5 ± 7.9	3.4 ± 1.7	3.7 ± 0.8	5.3 ± 2.1	0.2		
Amount FAOH (g/100 g sample weight)	4.2 ± 1.2	0.03 ± 0.01	0.07 ± 0.02	0.02 ± 0.01	2.4 ± 1.0	0.04 ± 0.05		
14:0	0.34 ± 0.10 0.13–0.42 5.0 ± 0.5	1.05 ± 0.62 0.34–1.65 7.1 ± 0.6	0.19 ± 0.11 0.04–0.29 5.1 ± 1.0	0.10 ± 0.06 0.04–0.16 2.4 ± 0.8	0.17 ± 0.04 0.11–0.20 3.3 ±0.8	0.006 0.004–0.008 3.0	0.14 0.09–0.21	7.3 ± 1.3
16:0	0.39 ± 0.09 0.21–0.47 5.9 ± 0.5	2.29 ± 1.23 1.00–3.39 16.1 ± 2.0	0.52 ± 0.25 0.16–0.72 15.2 ± 0.4	0.59 ± 0.29 0.38–0.92 15.9 ± 0.1	0.47 ± 0.16 0.27–0.66 9.0 ± 0.8	0.018 0.013–0.023 9.6	0.50 0.37–0.79	15.8 ±2.6
Sum SFA	0.90 ± 0.21 0.73–1.10 13.0 ± 1.5	3.78 ± 2.04 1.58–5.52 26.5 ± 2.4	0.85 ± 0.43 1.10–4.88 24.7 ± 1.1	0.86 ± 0.41 0.53–1.31 23.0 ± 0.8	0.74 ± 0.22 0.45–0.99 14.5 ± 1.8	0.035 0.03–0.04 18.5		26.9 ± 4.8
18:1n-9	1.35± 0.43 0.62–1.82 19.8 ± 3.1	1.35 ± 0.79 0.57–2.207 9.4 ± 1.8	0.43 ± 0.19 0.16–0.509 13.4 ± 3.5	0.80 ± 0.36 0.57–1.22 21.8 ± 3.5	0.80 ± 0.43 0.50–1.43 15.1 ± 4.0	0.029 0.025–0.033 16.6	0.44 0.29–0.76	10.0 ± 3.3
20:1n-9	0.53 ± 0.17 0.17–0.73 7.6 ± 1.1	1.52 ± 0.89 0.36–2.216 10.0 ± 2.2	0.24 ± 0.18 0.01–0.43 5.9 ± 3.2	0.19 ± 0.13 0.09–0.34 4.7 ± 1.2	0.55 ± 0.26 0.18–0.77 9.9 ± 2.3	0.018 ± 0.005 0.015–0.022 9.7	0.21 0.10–0.42	5.4 ± 4.8
20:1n-11	0.13 ± 0.4 0.04 -0.17 3.0 ± 0.8	0.15 ± 0.08 0.13–0.23 1.9 ± 0.3	0.03 ± 0.02 0.00–0.04 1.0 ±0.2	0.04 ± 0.03 0.02–0.07 0.7 ± 0.2	0.17 ± 0.09 0.05–0.25 1.0 ± 0.2	0.00 0.002–0.003 1.4	0.045 0.02–0.09	0.5 ± 0.4
22:1n-11	0.78 ± 0.24 0.26–1.07 11.3 ± 1.5	3.08 ± 1.77 0.74–4.28 20.4 ± 4.8	0.26 ± 0.22 0.01–0.54 6.1 ± 4.15	0.20 ± 0.16 0.09–0.38 4.9 ± 1.5	0.52 ± 0.23 0.19–0.69 9.5 ±2.1	0.027 0.019–0.035 14.4	0.28 0.11–0.67	7.5 ± 7.0
Sum MUFA	4.03 ± 1.14 1.60–5.05 57.1 ± 5.8	7.82 ± 4.41 2.40–11.84 52.7 ±5.8	1.42 ± 0.79 0.34–2.21 39.6 ± 6.6	1.75 ± 0.95 1.12–2.84 46.1 ± 4.1	3.05 ± 1.33 1.33–4.58 56.0 ± 4.1	0.104 0.09–0.11 56.7		35.5 ±12.8
18:2n-6	0.12 ± 0.03 0.06–0.15 1.8 ± 0.2	0.19 ± 0.10 0.07–0.28 1.2 ± 0.1	0.08 ± 0.03 0.04–0.11 1.7 ± 0.4	0.06 ± 0.02 0.04–0.07 1.7 ± 0.4	0.11 ± 0.05 0.07–0.18 2.2 ± 0.4	0.002 0.002–0.003 1.2	0.04 0.03–0.07	1.6 ±0.4
20:4n-6	0.04 ± 0.01 0.02–0.05 0.6 ± 0.1	0.05 ± 0.02 0.03–0.06 0.4 ± 0.1	0.03 ± 0.01 0.03–0.04 1.2 ± 0.7	0.04 ± 0.01 0.03–0.04 1.1 ± 0.4	0.04 ± 0.01 0.03–0.05 0.7 ± 0.2	0.002 0.002–0.002 1.0	0.03 0.01–0.04	0.9 ±0.4
SUM n-6	0.217 ± 0.050 0.12–0.26 3.2 ± 0.3	0.31 ± 0.15 0.14–0.46 2.3 ± 0.2	0.15 ± 0.05 0.08–0.19 5.0 ± 1.9	0.13 ± 0.03 0.10–0.16 3.7 ± 0.8	0.20 ± 0.08 0.13–0.30 3.8 ± 0.7	0.01 0.01–0.01 6.6		2.8 ± 0.5
20:5n-3	0.414 ± 0.104 0.21–0.51 6.2 ± 0.6	0.61 ± 0.35 0.28–0.97 4.3 ± 0.8	0.32 ± 0.14 0.15–0.44 10.2 ± 2.7	0.37 ± 0.14 0.25–0.52 10.3 ± 1.1	0.46 ± 0.16 0.27 - 0.65 8.9 ± 0.8	0.013 0.008–0.017 2.6	0.23 0.15–0.36	11.8 ± 4.1
22:6n-3	0.68 ± 0.13 0.42–0.79 10.4 ± 1.6	1.11 ± 0.57 0.61–1.79 8.2 ± 2. 5	0.45 ± 0.19 0.21–0.61 14.2 ± 3.1	0.44 ± 0.16 0.30–0.62 12.1 ± 1.2	0.47 ± 0.16 0.29–0.68 9.1 ± 1.2	0.05 0.04–0.06 6.6	0.53 0.44–0.64	10.2 ± 2.2
Sum n-3	1.54 ± 0.35 0.84–1.84 22.5 ± 3.1	2.44 ± 1.31 1.16–3.97 17.5 ± 3.5	0.96 ± 0.44 0.41–1.33 29.7 ± 5.7	0.96 ± 0.37 0.61–1.36 26.4 ± 3.0	1.27 ± 0.47 0.72–1.86 24.3 ± 1.9	0.039 0.03–0.05 3.3	0.92 0.71–1.25	30.1 ± 6.9
Sum PUFA	1.85 ± 0.41 1.00–2.22 27.1 ± 3.3	2.89 ± 1.52 1.35–4.62 20.8 ± 3.7	1.14 ± 0.49 0.50–1.55 35.6 ± 7.2	1.12 ± 0.42 0.73–1.56 30.9 ± 3.6	1.54 ± 0.57 0.88–2.26 29.5 ± 2.5	0.047 0.03–0.06 24.9		33.7 ± 7.6

* Measurements on individual samples, data from https://sjomatdata.hi.no/#search/. ** Fish oil used for feed production [39].

**Table 5 foods-09-00344-t005:** Potential contribution (%) of selected micronutrients to recommended intake (RI) in women from a serving of 50 g mesopelagic species in comparison to cod, salmon, sprat, beef, pork and chicken, and the potential doses of daily RI for women per km^3^ fjord.

Potential contribution to RI (%)	Iodine	Calcium	Iron	Zinc	Selenium	Vitamin A1	Vitamin D3
RI ^a^	150 µg	800 mg	15 mg	7 mg	50 µg	700 RE	10 µg
*Benthosema glaciale*	14	31	4	6	61	117	<LOQ
*Maurolicus muelleri*	9	34	5	8	44	73	<LOQ
*M. norvegica*	40	41	7	7	101	5	<LOQ
*Pasiphaea sp.*	15	40	1	6	43	1	<LOQ
*Eusergestes arcticus*	39	33	1	13	52	3	<LOQ
*Periphylla periphylla*	1	3	0	8	4	0	<LOQ
Salmon filet (*Salmo salar)*	1	0	1	3	17	-	43
Cod filet *(Gadus morhua)*	63 *	-	0	3	25	1	-
Sprat *(Sprattus sprattus)*	2	-	6	15	36	-	-
Pork	0	0	3	13	6	0	0
Chicken ^b^	0	1	2	11	12	1	0
Beef	1	0	8	29	6	0	0
No of daily doses of RI from mesopelagic species/ km^3^ fjord ^c^	169,000	353,000	31,800	87,300	591,000	348,000	-

RE = retinol equivalent; ^a^ RI: recommended intake, values according to [39]. Values for *Salmo salar* and *Gadus morhua* are from the seafood database (https://sjomatdata.hi.no/#search/). No data on calcium and vitamin D in *Gadus morhua*, and vitamin A1 in *Salmo salar*, in the seafood database. The values in *Salmo salar* represents farmed salmon. Value from pork, chicken and beef from the Norwegian food database (https://www.matvaretabellen.no/?language=en). For vitamin A, 1µg retinol (A1) = 1 retinol equivalent (RE). LOQ for vitamin D < 1 µg. ^b^ Chicken, thigh, no skin, raw. ^c^ calculated from [39]. * Value from [40].

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
