# Peer review of "Mesopelagic Species and Their Potential Contribution to Food and Feed Security—A Case Study from Norway"

_foods, 2020, doi:10.3390/foods9030344_

Round 1

Reviewer 1 Report

The manuscript has evaluated nutritional composition and value of some mesopelagic marine species. It provides very interesting information and can be improved after some revision according the following comments.

Line 14; The full scientific name of the two fish species should be mentioned in the abstract as their first appearance.

Line 104-105; The used conversion factor (6.25) cannot be correct for both crustacean and vertebrates. In case of crustacean, chitin can also contribute to a large extent to the measured amount of nitrogen which needs to take into account during the calculation of protein content using nitrogen content. This must be reformed and clarified.  

Please use the correct conversion factor for each species. Please use the following reference:

Mariotti F, Tomé D, Mirand PP. Converting nitrogen into protein - Beyond 6.25 and Jones’ factors. Crit Rev Food Sci Nutr 2008;48:177–84. https://doi.org/10.1080/10408390701279749.

Line 182-236; In all reported result tables, it should be clarified if the reported nutrient contents are wet weight basis or dry weight basis.

Please include possible seasonal variation in the composition of the studied species in your discussions and conclusion. 

Author Response

Line 14 --> The full scientific name of the two fish species has been added.

Line 104-105; The used conversion factor (6.25) cannot be correct for both crustacean and vertebrates. In case of crustacean, chitin can also contribute to a large extent to the measured amount of nitrogen which needs to take into account during the calculation of protein content using nitrogen content. This must be reformed and clarified.
--> We acknowledge this comment and are aware of the issue. However, we did not analyze our samples for amino acids and do not have a scientifically sound basis, especially for the shrimp and jellyfish, to calculate the true protein content. To make our result comparable to most of the existing literature and references (food composition databases including https://sjomatdata.hi.no/#search/), we chose to use the factor of 6.25.
We have added a clarifying sentence in the text:"Since the here applied crude protein method is somewhat uncertain due to the assumption that all measured nitrogen comes from protein and a standard protein composition, in future studies, the amino acid profile should be considered for calculating true protein content [91,92]".

Line 182-236; In all reported result tables, it should be clarified if the reported nutrient contents are wet weight basis or dry weight basis
--> The information was included in the footnote except. Now it has been moved to the caption to make it more visible.

Please include possible seasonal variation in the composition of the studied species in your discussions and conclusion.
-->More details were added to the discussion. However, as season has not been shown to be important, we decided to not include this issue in the conclusion.

Reviewer 2 Report

Anita R. Alvheim and colleagues provide a detailed biochemical characterization of six mesopelagic species from Norwegian fjords in their work entitled “Mesopelagic species and their potential contribution 2 to food and feed security – a case study from Norway”.

The study is certainly a relevant and timely contribution for the scope of a publication as Foods, making available valuable data for those working on the nutritional aspects of seafood for human consumption and as feed ingredients (mostly for marine aquaculture).

My main concerns with the manuscript on its present form is the discussion that the authors have made on their findings assuming that these species will be consumed whole as a food. It is very unlikely that consumers will accept fish that has not been de-gutted nor de-headed, as most countries currently offer consumers fish fillets (not even whole fish are often available in super-markets in developed countries). The same is valid for shrimp, as these are commonly peeled and de-headed… As such, the claims made by the authors on the nutritional value of the species they have surveyed may be overestimated, namely for their mineral content (due to fish bones and shrimp exoskeleton), as well as for their fatty acid content (due to the presence of fish liver and shrimp digestive gland in homogenized samples). This issue is not valid for their use as feed, hence the discussion performed being valid for this specific use.

The authors must also question if these fisheries will be sustainable. History advise us not to be so optimistic when claiming that “there is too much biomass, we will not be able to overfish”. This was the mindset in the late 19th/early 20th century when discussing cod-fish stocks and look at them now… The authors must emphasize this challenge, as well as the occurrence of potential seasonal and interannual shifts in the biochemical composition of mesopelagic species (the present work addressed samples collected in December, will species collected during late spring/summer display similar profiles?).

I have made a number of comments on the PDF file I am attaching and I look forward to see the revised version of this study.

I congratulate the authors for their effort and timely contribution.

Author Response

My main concerns with the manuscript on its present form is the discussion that the authors have made on their findings assuming that these species will be consumed whole as a food. It is very unlikely that consumers will accept fish that has not been de-gutted nor de-headed, as most countries currently offer consumers fish fillets (not even whole fish are often available in super-markets in developed countries). The same is valid for shrimp, as these are commonly peeled and de-headed… As such, the claims made by the authors on the nutritional value of the species they have surveyed may be overestimated, namely for their mineral content (due to fish bones and shrimp exoskeleton), as well as for their fatty acid content (due to the presence of fish liver and shrimp digestive gland in homogenized samples). This issue is not valid for their use as feed, hence the discussion performed being valid for this specific use.
-->Regarding this we are quite certain, that especially the studied fish species will be consumed as a whole. With a length below seven cm, these fish might be among the smallest directly consumed fish. However, even for larger species like sardines, it is not uncommon to be consumed as whole. Especially if we leave the most developed countries, it is quite common to eat whole fish. And it is maybe there, the biggest need for mesopelagic fish as food lies. It is reported that the mesopelagic biomass is delicate when fished and brought onboard and we doubt that an extensive processing like gutting and even filleting is realistic. Already the size would the processing make expensive and cumbersome. The investigated shrimp species are also small and do not have large muscular tails like the commercially fished shrimps. Peeling these shrimps would be cumbersome and inefficient. This especially applies to krill. Some of species have been tried whole by a local renown chef who definitely approved their compatibility as whole eaten food if they find there way through legislation. 
For clarification we added: "Small fish consumed whole including head and viscera is already part of multiple food cultures [51,86,87], which probably also will apply to most mesopelagic species being of small size"
In L.315ff we also state that "All data presented here are from whole, raw specimens. Processing might affect the amount of micronutrients [54,55]. Thus, further studies are needed to investigate processing on micronutrient content in the mesopelagic species presented here."

The authors must also question if these fisheries will be sustainable. History advise us not to be so optimistic when claiming that “there is too much biomass, we will not be able to overfish”. This was the mindset in the late 19th/early 20th century when discussing cod-fish stocks and look at them now…  
--> We did not consider this discussion as within the main scope of the paper and therefore only brought it to the attention of the reader in the concluding section. The here presented data is not suited for a scientifically based discussion on sustaniability. However, the issue of a sustainable fisheries has now in addition been added to L.51 and L.388

The authors must emphasize this challenge, as well as the occurrence of potential seasonal and interannual shifts in the biochemical composition of mesopelagic species (the present work addressed samples collected in December, will species collected during late spring/summer display similar profiles?).
--> We are only aware of one study actually addressing this issue and the main findings have been explained. L.408

Reviewer 3 Report

The manuscript Mesopelagic species and their potential contribution to food and feed security – a case study from Norway was very interesting to read. It was well written and presented, and the research is undoubtedly both important and timely. I would also like to complement the modest approach the authors take with their paper: contribute with novel data…. And also the conclusion: potentially….

I have commented upon some minor details and issues that may improve the manuscript. The reference list should be revised as some references are incomplete, some do not included the last page number and some page numbers are listed with an s. Also in the text, some references come directly after the word, while others come with a space between.

Line 26: I think one should not start a sentence with a number. Suggestion: As many as 842 million people

Line 86: Is this correct, or should it be 2018?

Line 165: technical flaws in the equation

Line 183: This first sentence is a repetition of the introduction and aim. The second sentence would be more suitable in the discussion part.

Line 225, “We” should be excluded from the result section.

Table 4. The latin name of blue whiting should be included

Section 3.3 and 3.4 could be included in the discussion section instead of the result section.

Line 271: The first sentence here as well is a repetition from the introduction. It should be rephrased for a better flow.

Line 352: “..nutrient. It…”

Line 407: Also here, the sentence appears as a repetition. This sentence could be rephrased to “in this study six mesopelagic species from tree fjords in Western Norway were investigated for nutritional composition and evaluated as potential for food and feed security”.

Line 408: The sentence “Small fish eaten whole has a potential….” Is redundant in this conclusion.

Author Response

The reference list should be revised as some references are incomplete, some do not included the last page number and some page numbers are listed with an s. Also in the text, some references come directly after the word, while others come with a space between.--> Reference list has been checked and errors fixed

Line 26: I think one should not start a sentence with a number. Suggestion: As many as 842 million people--> has been changed

Line 86: Is this correct, or should it be 2018? --> Thanks, has been changed to 2018

Line 165: technical flaws in the equation --> Has been addressed, please see manuscript.

Line 183: This first sentence is a repetition of the introduction and aim. The second sentence would be more suitable in the discussion part. -->Sentence one was rephrased. As reviewer 2 requested a clear statement why blue whiting was used in the tables, we prefer to leave the sentence.

Line 225, “We” should be excluded from the result section. --> Has been changed

Table 4. The latin name of blue whiting should be included --> Has been included

Section 3.3 and 3.4 could be included in the discussion section instead of the result section. --> The presented data here is directly based on our results and we choose to present it here to have a tidy outline, although it also would fit into the discussion.

Line 271: The first sentence here as well is a repetition from the introduction. It should be rephrased for a better flow. --> Has been rephrased to: Here, the nutrient composition of six mesopelagic species from Norwegian fjords  is discussed and their contribution to global food and feed security evaluated."

Line 352: “..nutrient. It…” -->Has been inserted.

Line 407: Also here, the sentence appears as a repetition. This sentence could be rephrased to “in this study six mesopelagic species from tree fjords in Western Norway were investigated for nutritional composition and evaluated as potential for food and feed security”.
--> Has been changed to: "In this study six mesopelagic species from three fjords in Western Norway were investigated for nutritional composition and evaluated in terms of food and feed security"

Line 408: The sentence “Small fish eaten whole has a potential….” Is redundant in this conclusion.--> Reviewer 2 strongly emphasizes that we were analyzing whole fish, and raises doubts that this will be the case. We therefor would like to leave this phrase here to make sure for the reader that we looked at whole fish, which we think is reasonable.

Round 2

Reviewer 2 Report

I congratulate the authors for successfully addressing all issues that I have pointed on the earlier version of their manuscript.